# IL-10 and IL-17 as Progression Markers of Syphilis in People Living with HIV: A Systematic Review

**DOI:** 10.3390/biom12101472

**Published:** 2022-10-13

**Authors:** Adriana Hernández-Pliego, Dayana Nicté Vergara-Ortega, Antonia Herrera-Ortíz, Cairo Toledano-Jaimes, Fernando R. Esquivel-Guadarrama, Miguel Ángel Sánchez-Alemán

**Affiliations:** 1Centro de Investigación Sobre Enfermedades Infecciosas, Instituto Nacional de Salud Pública, Cuernavaca 62100, Mexico; 2Facultad de Farmacia, Universidad Autónoma del Estado de Morelos, Cuernavaca 62210, Mexico; 3Facultad de Medicina, Universidad Autónoma del Estado de Morelos, Cuernavaca 62350, Mexico

**Keywords:** Syphilis, HIV, people living with HIV (PLWH), immune response, co-infection

## Abstract

Much is known about the natural history of syphilis; however, less is known about the immune response against it, and even less is known about people living with HIV (PLWH). Due to the lack of an animal model to study host-pathogen interactions, it remains unclear how the host eliminates the bacteria. Here, we attempt to elucidate the immune response against infection by summarizing all the reported data in a systematic review. We found that only seven papers included PLWH, and they did not accurately describe the immune response against *Treponema pallidum* since only lymphopenia was reported upon coinfection. On the other hand, at least sixteen papers described the host-pathogen interaction in individual cell populations. Using this information, we established the kinetics of the immune response against syphilis and hypothesized how CD4^+^ T cells, such as Th17 and T rex cells, worsen the progression of the disease in PLWH through their hallmark cytokines, IL-10 and IL-17, and how these two cytokines may play important roles as biomarkers.

## 1. Introduction

Syphilis is a sexually transmitted disease (STD) caused by *Treponema pallidum pallidum* (*Tpp*). It was considered to be eradicated in the 1990s, but it remerged in 2000, especially among men who have sex with men (MSM) and people living with HIV (PLWH). Although penicillin is available worldwide and is the first line of treatment, the number of syphilis cases has been increasing. In 2020, the World Health Organization (WHO) estimated 7 million new syphilis cases [1].

When an infected person has sexual intercourse with a healthy person, the pathogenic period begins, wherein inoculated bacteria begin to replicate and disseminate extracellularly. Within the first 3 to 90 days after inoculation, clinical manifestations such as lymphadenopathy and a single painless chancre may occur, called primary syphilis (PS). PS can self-resolve and go through a latent phase, which can last for months, and then reactivate again, showing clinical manifestations such as systemic maculopapular rash. This stage is known as secondary syphilis (SS). Again, the symptoms diminish a few days and years later (ten to twenty years) appear again in severe ways, such as cardiopathy or neurological damage [2,3]. Tertiary syphilis (TS) is known as the last phase, which can end in death if not treated opportunely.

In PLWH, the primary and secondary stages of syphilis overlap, presenting multiple chancres and rashes at the same time, making it more complex to diagnose a single stage. Even worse, the progression to a tertiary stage becomes faster with HIV coinfection, and it is still unknown how this may be possible with an undetectable viral load controlled with antiretroviral treatment [3]. The immune system plays a relevant role in the progression of both diseases; however, an important question remains: what CD4^+^ T cell population interacts with *Tpp* when patients are immunocompromised by HIV?

Due to the lack of an animal model to study the interactions between the host and pathogen, it remains unclear how the host eliminates the bacteria and how it mimics other diseases. Here, we attempt to elucidate the role of the CD4^+^ T cell response against infection with *Tpp* in PLWH.

## 2. Materials and Methods

The aim of this review is to clarify and understand the CD4^+^ T cell population against syphilis, specifically in PLWH. We followed the COSMOS-E guidelines for Systematic Reviews for Observational Studies of the Etiology of the disease [4]. This systematic review was also registered in the PROSPERO system under the ID CRD42022361626 (under review). Ten research criteria were used in four different databases (PubMed, Redalyc, Cochrane DSR and Cochrane RCT). This research began in June 2018 and ended in January 2022. The most significant results were “*Treponema pallidum*” AND “HIV” , “Syphilis” AND “immune response” , “HIV immune response” , and “Syphilis HIV immunology” (Figure 1). The most relevant articles were found within the following research criteria:(“treponema pallidum” [MeSH Terms] OR (“treponema” [All Fields] AND “pallidum” [All Fields]) OR “treponema pallidum” [All Fields]) AND (“immunologie” [All Fields] OR “immunology” [MeSH Subheading] OR “immunology” [All Fields] OR “allergy and immunology” [MeSH Terms] OR (“allergy” [All Fields] AND “immunology” [All Fields]) OR “allergy and immunology” [All Fields] OR “immunology” [All Fields])(“syphilis” [MeSH Terms] OR “syphilis” [All Fields]) AND (“hiv” [MeSH Terms] OR “hiv” [All Fields]) AND (“immunologie” [All Fields] OR “immunology” [MeSH Subheading] OR “immunology” [All Fields] OR “allergy and immunology” [MeSH Terms] OR (“allergy” [All Fields] AND “immunology” [All Fields]) OR “allergy and immunology” [All Fields] OR “immunology s” [All Fields])(“cells” [MeSH Terms] OR “cells” [All Fields] OR “cellular” [All Fields]) AND (“response” [All Fields] OR “responses” [All Fields] OR “responsive” [All Fields] OR “responsiveness” [All Fields] OR “responsivenesses” [All Fields] OR “responsives” [All Fields] OR “responsivities” [All Fields] OR “responsivity” [All Fields]) AND (“syphilis” [MeSH Terms] OR “syphilis” [All Fields])(“humoral” [All Fields] OR “humorally” [All Fields] OR “humoral” [All Fields]) AND (“response” [All Fields] OR “responses” [All Fields] OR “responsive” [All Fields] OR “responsiveness” [All Fields] OR “responsivenesses” [All Fields] OR “responsives” [All Fields] OR “responsivities” [All Fields] OR “responsivity” [All Fields]) AND (“syphilis” [MeSH Terms] OR “syphilis” [All Fields])

## 3. Results

The outcomes from this systematic review were 184 papers of interest. From this information, we chose the most relevant for this review. One hundred and sixty-six papers were discarded because the information was not from *Treponema pallidum* subs *pallidum*. We analyzed 18 papers in detail, including sample size; type of sample; human and/or animal studies; type of study; genre and language; and cell type response against *Tpp*, such as CD4^+^ T cell lymphocytes and their subtypes, Th1, Th2, Trex, Th17, Th22. Studies published in any language other than English or Spanish were excluded. Duplicated records were eliminated. Two authors (AHP and MASA) screened the abstract of each article, and full texts deemed relevant were reviewed to reach a consensus for inclusion or exclusion. Reports of individual patient data were excluded. Of the eighteen papers reviewed, only eleven were exclusively about *Tpp* (see Appendix A).

### 3.1. Innate Immune Response against Treponema pallidum

Cruz et al. tested skin lesions in SS patients and reported high levels of natural killer (NK) cells, suggesting that there is an innate response in the damaged area, even though it is not a new infection [5]. This suggests the involvement of trained immunity, and we can speculate about the resolution of the disease in advanced stages due to the innate response. Trained immunity is usually referred to as the epigenetic reprogramming of transcriptional pathways instead of genetic recombination [6,7].

In recent years, researchers have reported that innate immunity has been evolving and developing memory as well as adaptive immunity [8]. Trained immunity lasts at least one year and can help to eliminate infectious diseases, although it is not as specific as the adaptive response. Nevertheless, it has been reported that, in chronic inflammatory conditions, trained immunity may contribute to hyperinflammation and the progression of cardiovascular disease, autoinflammatory syndromes and neurological damage [7].

Trained immunity may be key to understanding Cupid’s disease (as neurosyphilis is known), but further research is needed. At the third phase of the disease, neurosyphilis (NS), NK cells remain high in the blood, suggesting that trained immunity is responding even months after the first encounter, although the spirochete still remains in the host. On the other hand, macrophages, as innate cells, phagocytose *Tpp*. Phagocytosis can occur in two ways: TLR1 and TLR2 can recognize lipoproteins on the outer membrane of the spirochete and facilitate the formation of a phagosome or antibodies (IgG or IgM) bound to these lipoproteins can be opsonized by recruited complement proteins, which helps the macrophages or neutrophils phagocytose the whole bacteria (see Figure 2) [9,10].

### 3.2. Adaptive Immune Response against Syphilis

While it is an extracellular bacterium with poor outer membrane proteins, *Treponema pallidum* may be recognized by any antigen-presenting cell (APC), such as dendritic cells (DCs) or macrophages. APCs capture and phagocytose the Tpp, migrate to the nearest lymph node and present antigens in the major histocompatibility complex (MHC) class II context to CD4^+^ T cells. In this immunological synapse, the APC, usually a DC, produces IL-12 to stimulate the polarization of Th cells into a proinflammatory Th1 profile. This subset of Th cells, after activating their canonical transcription factors T-bet and STAT1, commonly produces TNF and IFN-γ as the principal cytokines for a proinflammatory profile [11,12]. These last two cytokines have been reported in lesions from infected patients, suggesting that there is a proinflammatory localized response (See Figure 3).

In the systemic response to PS, high levels of CD4^+^ T cells have been reported, with higher concentrations of IFN-γ, TNF for Th1 subtype and IL-10 and TGFβ for Trex cells. These cytokines are necessary to induce the inflammasome. On the other hand, Th2 cells do not seem to play a significant role in the clearance or persistence of *Tpp* [5,12,13]. IFN-γ activates and stimulates macrophages for an aggressive response to clear the infection [8], but Th17 cells do not play a relevant role in the primary response. However, IL-17 is increased by CD8^+^ and NK cell responses in an attempt to recruit neutrophils to phagocytose the bacteria. A research group in China [14] recently discovered that *T. pallidum* delays neutrophil apoptosis through intrinsic and extrinsic pathways. *Tpp* stimulates the secretion of the anti-apoptotic cytokine IL-8 to prolong the neutrophil lifespan. Neutrophils do not play a role in the pathogenesis of the disease; however, this finding complements the panorama between the complex cellular interactions of the immune system. Although these data were obtained from in vitro assays, they could guide new observational studies, especially in immunosuppressed individuals, such as PLWH.

The next stage is secondary syphilis, where the levels of Th1 and Trex remain high in peripheral blood. Cruz et al. in 2012 found lower levels of DCs and NK cells in peripheral blood samples from secondary syphilis patients, suggesting tissue infiltration and cytotoxic activity from these two cell types. This research group also described the importance of opsonophagocytosis in ex vivo assays and cell-mediated immunity for bacterial clearance. They have also reported higher levels of macrophages undergoing diapedesis, indicating increased recruitment of cells to damaged areas and higher levels of molecules for processing antigens [10]. Both Wang and collaborators and Cruz and collaborators highlighted the role of antibody-mediated phagocytosis and the participation of both macrophages and neutrophils in the skin as local responses against spirochetes [5,10,14,15].

In the tertiary stage of the natural history of the disease, the worst damage begins in the central nervous system (CNS) and cardiovascular system (CVS). The damage spreads through multiple organ failure. Eventually, people with TS die. The presence of IgM and IgG in cerebrospinal fluid (CSF) has been described, which suggests that the infection has broken all barriers and disseminated to the brain [10,11]. Interestingly, at this point of the disease, both CD8^+^ T and Th17 cells have increased levels. It is unclear how they might be acting to diminish the damage or if they might be acting as self-enemies. Th17 cells are effector cells that respond to extracellular bacteria and are present in autoimmune diseases [16,17,18]. Syphilis has been described to disguise as other autoimmune illnesses, such as lupus, multiple sclerosis, psoriasis or tabes dorsalis, among others [19]. Thus, an accurate diagnosis is often confused, and the opportunity for a specific treatment against the stealthy spirochete is eliminated. This last point is critical, and the Th17 cell may be the key to understanding why *Tpp* is a mimic and stealthy disease.

### 3.3. Co-Infection: HIV/Syphilis

The problem of understanding how the immune system responds to syphilis becomes larger when considering HIV coinfection. HIV seems to orchestrate the immune response for many possible reasons, one of which could be that being the first infection in the organism gives it the capacity to control the immune response. On the other hand, acquiring syphilis before HIV increases the probability of acquiring the viral infection by 2.3 times [11].

We have been studying the HIV immune response since its first appearance in the 1980 pandemic. Even though syphilis was the first STI pandemic in the world (according to some researchers [20,21], mummies presented tabes dorsalis), we still do not fully understand its complexity because of the lack of an accurate animal model to mimic the natural history of the disease. According to a previous systematic review, we detected at least seven papers in which the immunological response to coinfection in the three different syphilis stages was discussed. Only two papers specifically included MSM.

In 2009, Knudsen et al. [22] reported a strong association between increased IL-10 levels and the primary stage of syphilis in HIV-positive patients, even though the CD4^+^ T cell response was low, and this cytokine, along with TNF, diminished after antibiotic treatment. Knudsen and colleagues based their study on serum samples of 36 patients with HIV. During and after syphilis infection, they were tested for different cytokines, where they detected IL-2, IL-4, IL-6, IL-8, IL-10, IFN-γ and TNF. Above all, IL-10 was expanded significantly, with levels of 12.8 pg/mL to 46.7 pg/mL at the time of syphilis diagnosis, which decreased to 13 pg/mL after antibiotic treatment.

Years later, in 2016, Kotsafti and colleagues [23] observed a decrease in CD4^+^ T cells in 85% and 80% of people who took antiretroviral therapy and those who did not, respectively, in primary syphilis. After syphilis treatment, CD4^+^ T cells returned to “normal” levels. The aim of Kotsafti’s retrospective research was to evaluate whether there was any modification for the HIV markers during syphilis coinfection. He found that early syphilis worsens the progression of HIV infection. However, what makes them worse?

Other research groups began to study the link between innate and adaptive immune responses. In 2017, Li et al. [24] investigated gamma delta T cells (Tɣδ). They recruited a cohort of 57 PLWH in Beijing. Within this cohort, 33 were coinfected with syphilis. They analyzed Tɣδ cells in serum using flow cytometry to determine their vestigial phenotype. Li et al. reported that syphilis coinfection may revert the disequilibrium of Vδ1 and Vδ2 in acute HIV infections. In addition, *Tpp* infection can diminish the activation of T gamma delta cells in chronic HIV infection. Chronic HIV-infected people were demonstrated to have more IL-17 produced by Tɣδ cells than those who had acute HIV infection, regardless of syphilis stage. These increased levels of IL-17 correlated positively with neutrophil percentage, suggesting that they may play a role in the progression of HIV and that they may be a link between innate and adaptive immunity using trained immunity to face the spirochete in chronic viral diseases.

In 2019, Guo et al. [25], within the same group as Li, demonstrated that coinfection changes monocyte subsets and Trex using flow cytometry. They enrolled 81 participants and assigned them to different groups: syphilis-positive (RPR+), chronic HIV infection (CHI), syphilis/HIV (RPR+/CHI) and the reference group (Ctrl). The latter lacked any opportunistic infection, tuberculosis, hepatitis B or C virus. Using peripheral blood samples, they isolated mononuclear cells, cultivated them and stained them for flow cytometry. They reported that intermediate monocytes (CD14^++^CD16^+^) control the differentiation of Trex subsets in Tpp/HIV coinfection.

One year later and using the same population [26], in 2020, they described how CCR2 and CX3CR1 (rolling and chemokine molecules) increased their density in the three different subsets of monocytes, regardless of retroviral therapy. These data suggest that there is more diapedesis in those who are coinfected than in those who do not suffer from any infection and are considered healthy controls, which is due to the migration and presentation of antigens in the nearest lymph nodes.

For primary/secondary syphilis, there was only one study from Kenyon et al. in 2017 [27]. They retrospectively studied 79 people living with HIV and 12 without HIV, both groups with a new diagnosis of syphilis. Using peripheral blood samples, they detected a pool of cytokines (IFN-α, IFN-γ, IL-1β, IL-12p40, IL-12p70, IP-10, MCP-1, MIP-1α, MIP-1β, IL-4, IL-5, IL-6, IL-7, IL-8, IL-10 and IL-17a) before antibiotic treatment and six months later. They reported a nine times greater concentration of IL-10 in CD4^+^ T cells, and those levels became lower but did not return to baseline after treatment. This may suggest that T rex are playing an important role in coinfection, although this subset of T cells is known to be significant in the progression of HIV into AIDS, along with Th17, despite the use of antiretroviral therapy to compensate for the damage caused by the virus. It is important to remark that a delicate equilibrium is necessary for better health in coinfection. When these two subsets of T cells collide, autoimmunity occurs.

Regarding tertiary syphilis, there was only one study published since 2021. Yan et al. [28] performed a retrospective study to compare noninvasive predictors of neurosyphilis among a HIV cohort in China. They enrolled 528 patients, where 143 had syphilis and no HIV infection and 385 were coinfected (HIV and syphilis) patients. At least 304 neurosyphilis patients were identified as having HIV. They concluded that a high-serum titer (>1:64) in TRUST is a neurosyphilis predictor among all participants (HIV-positive and HIV-negative); being ≥50 years old in those who were HIV-negative is another predictor to develop neurosyphilis; and fewer than 300 CD4^+^ T cells/µL is a predictor of asymptomatic neurosyphilis in HIV-positive participants. Nevertheless, lymphopenia is also present in other coinfections in PLWH, such as HSV-2, influenza virus, cytomegalovirus (CMV) and tuberculosis (TB), among others [29].

Another research group in China [30] aimed to study the role of IL-17 in 103 neurosyphilis patients, where at least 33% had neurological damage involvement and attended a one-year follow-up to monitor the disease. They found significantly increased levels of IL-17 in peripheral blood from Th17 cells compared with healthy donors. Those who presented a symptomatic form of neurosyphilis had higher levels of IL-17 derived from Th17 in the CSF. These results suggest that IL-17 is involved in the tertiary stage of syphilis, causing more damage than clearing the bacteremia in the central nervous system.

One limitation to better understanding this disease is the high percentage of asymptomatic syphilis, according to González Domenech [31], and at least 35% of patients belong to this group. This is a major problem because the disease can be misdiagnosed or not even treated, and not all MSM may be attending their physician appointments and, therefore, do not even notice a problem. There are various limitations to studying the immune response against *Tpp* in HIV-positive people, including the described lack of an accurate animal model and in vitro assays. Until the research focus is redirected or given due importance, some questions will remain unsolved: what causes asymptomatic syphilis? Does the immune system act like a friend or foe in coinfected individuals? Why does the immune system fail to generate a response to *Tpp*, and how does it persist for so long without activating clinical signals?

## 4. Conclusions

It remains unclear how CD4^+^ T cells play an important role in the persistence or clearance of the disease in PLWH. We can speculate about the asymptomatic presentation by hypothesizing that T rex may suppress Th1 by not allowing sufficient IFN-γ or TNF to recruit APCs or to stimulate other cells and, therefore, not being able to generate lesions. Another interesting point of view is innate lymphoid cells (ILCs), which are not a new topic in immunology. These innate cells have been recently reported to share the exclusive hallmark transcription factors of Th cells, such as T-bet, STAT1, STAT2 and STAT5 [32]. However, phenotypically, they may be different. It may be possible they have a role in trained immunity, and shared transcription factors complicate the landscape of a single-cell population. However, further research is needed to determine their role.

While PLWH has decreased CD4^+^ T cell levels, this subset is derived from naïve T cells; therefore, memory T cells and effector T cells are dramatically diminished. In an attempt to compensate for this, CD8^+^ T cells may increase their levels along with B cells secreting the polyclonal antibody anti-HIV (IgG) to clear or diminish the infection among CD4^+^ T cells. On the other hand, when *Tpp* appears, the primary response to clear the infection and present symptoms appear to be Th1-dependent, but T rex also arises in an attempt to control the inflammatory cascade caused by the spirochete. Over time, these two subsets decline, but Th17 cells appear at the last stage of syphilis, suggesting that this may worsen the clinical picture (See Appendix A).

Another recent topic is the epigenetic changes that the pathogen may be inducing in immune cells, which can be considered a virulence factor for evading immunity and causing nerve damage [33]. Flow cytometry and other molecular tools, such as RT-PCR, can be easily performed in the future. Can these two T cell subsets and their hallmark cytokine secretion biomarkers predict the progression of both diseases? Further research is required to elucidate the immune response against this binomial infection.

## Figures and Tables

**Figure 1 biomolecules-12-01472-f001:**
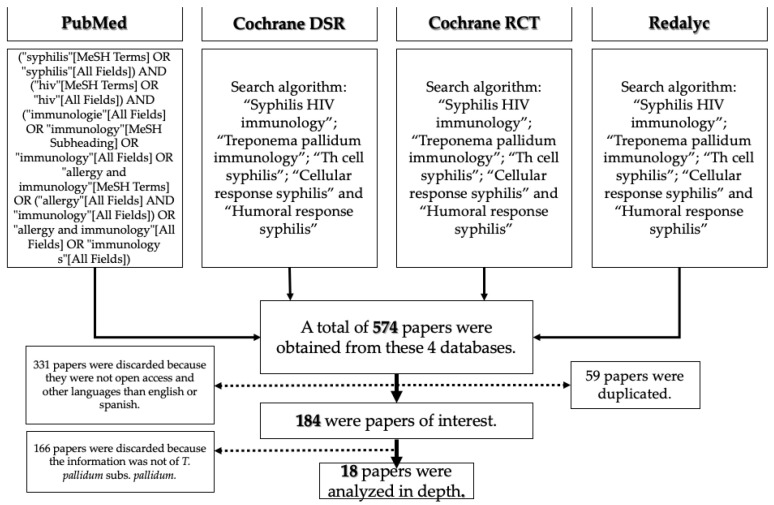
Election and methodology in four different databases for a systematic review.

**Figure 2 biomolecules-12-01472-f002:**
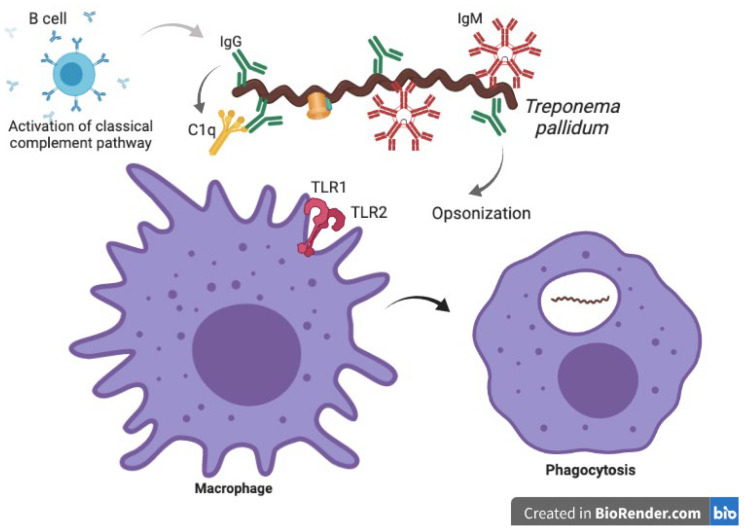
Innate immune response against *Treponema pallidum*. While *Tpp* is located extracellularly, B cells can secrete opsonizing antibodies and recruit complementing proteins to induce the classical pathway complement system. On the other hand, macrophages can recognize lipoproteins from the outer membrane of *Tpp* through TLR1 and TLR2 and phagocytose the bacteria.

**Figure 3 biomolecules-12-01472-f003:**
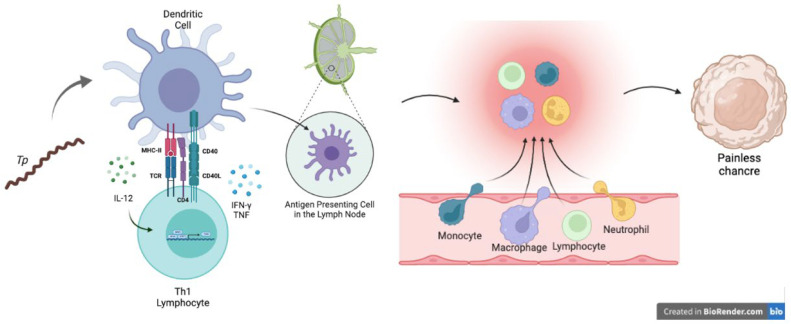
Immunological synergy between a dendritic cell and a T helper type 1 cell in response to Treponema pallidum and recruitment in the affected genital area. While Tpp is delivered into mucocutaneous wounds, DCs may recognize and phagocytose this pathogen, migrate toward lymph nodes and present antigen through MHC class II to Th lymphocytes. DCs begin to produce IL-12 to stimulate T cells to differentiate into the Th1 type and produce effector cytokines, such as TNF and IFN-γ. After antigen presentation, cells migrate from the lymph node to the affected area as a local response to clear the damage made by the bacteria.

## Data Availability

Not applicable.

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
