# Peer review of "IL-10 and IL-17 as Progression Markers of Syphilis in People Living with HIV: A Systematic Review"

_biomolecules, 2022, doi:10.3390/biom12101472_

Round 1

Reviewer 1 Report

Comment 1: The reference formatting is completely messed up in the manuscript.

Comment 2: There are some article usage problems throughout the manuscript. There are several grammatical mistakes.

Comment 3: The English language used in this manuscript is very poor. The author should rewrite some section to make it more appealing to the readers.

Comment 4: It is a Review article, the authors should not use section like Material and methods, results

Comment 5: Figure 1 is not cited in the text. What is the point of adding figure 1? It does not give any information to the reader. Instead, the authors can make a figure indicating the timeline for progress in the field.

Comment 6: The purpose of writing review is to assimilate information from other published article, organize the information and give a complete understanding of the past research. It not about just stating the past research

Comment 7: There is no mention of IL-10 and IL-17 in the introduction section

Comment 8: The authors can include some information about the treatment of syphilis

Comment 9: The authors can make the fonts bigger in Figure 2 and 3

Author Response

Comment 1: The reference formatting is completely messed up in the manuscript. 

The author MDPI guidelines have been revised, and references have been adjusted.

Comment 2: There are some article usage problems throughout the manuscript. There are several grammatical mistakes. 

The whole document has been sent to American Journal Experts to revise grammatical mistakes and correct style. The receipt is attached.

Comment 3: The English language used in this manuscript is very poor. The author should rewrite some section to make it more appealing to the readers. 

The whole document has been sent to American Journal Experts to revise grammatical mistakes and correct style.

Comment 4: It is a Review article, the authors should not use section like Material and methods, results 

In systematic reviews, the material and methods and the study registration number from the PROSPERO system are included. This allows other authors to replicate our research. The Biomolecules journal requests this section to publish any systematic review.

Comment 5: Figure 1 is not cited in the text. What is the point of adding figure 1? It does not give any information to the reader. Instead, the authors can make a figure indicating the timeline for progress in the field. 

Thank you for the comment. It has been revised and included in the text (Line 64).

Comment 6: The purpose of writing review is to assimilate information from other published article, organize the information and give a complete understanding of the past research. It not about just stating the past research 

We agree with the reviewer. The aim of the present systematic review was to combine information about the immune response against Treponema pallidum in people living with HIV and to propose the inflammatory features of the different biomarkers. This review does not only describe previous studies. Figures 1-3 are our own summary, and we had to assimilate the information to generate this figure. We believe that it summarizes our findings.

Comment 7: There is no mention of IL-10 and IL-17 in the introduction section 

The introduction provides a general perspective about the immune response against Treponema pallidum, in which IL10 and IL17 are not the highlight. That is why they are not mentioned in this section. When we first began this systematic review, we did not know which molecules were involved in the progression of syphilis in people living with HIV. As a result of the systematic research, we found that IL10 and IL17 were key markers of progression in coinfection, which is mentioned in the results section.

Comment 8: The authors can include some information about the treatment of syphilis 

We agree and have included a sentence on the treatment of syphilis in Lines 31 and 32.

Comment 9: The authors can make the fonts bigger in Figure 2 and 3 

We agree. Thank you for the recommendation. We have increased the font size (Lines 168-172 for Figure 2 and 249-253 for Figure 3).

Reviewer 2 Report

A very interesting and detailed review devoted to the study of the role of certain cytokines (IL-10, IL-17) in the progression of syphilis in patients with HIV infection. The possibility of using these cytokines as markers has been proven. The review is of considerable interest not only for infectious disease specialists working with HIV-infected patients, but also for classical immunologists. The review is written in good English and contains a number of interesting materials and conclusions. The review can be printed without significant changes.

Author Response

A very interesting and detailed review devoted to the study of the role of certain cytokines (IL-10, IL-17) in the progression of syphilis in patients with HIV infection. The possibility of using these cytokines as markers has been proven. The review is of considerable interest not only for infectious disease specialists working with HIV-infected patients, but also for classical immunologists. The review is written in good English and contains a number of interesting materials and conclusions. The review can be printed without significant changes.

Thank you for your comments. 

The manuscript has been revised by American Journal Experts to improve the paper.